# Investigating the Side-Effects of Neem-Derived Pesticides on Commercial Entomopathogenic and Slug-Parasitic Nematode Products Under Laboratory Conditions

**DOI:** 10.3390/plants8080281

**Published:** 2019-08-12

**Authors:** Renáta Petrikovszki, Pratik Doshi, György Turóczi, Ferenc Tóth, Péter Nagy

**Affiliations:** 1Plant Protection Institute, Faculty of Agricultural and Environmental Sciences, Szent István University, H-2100, Gödöllő, Hungary; 2Department of Zoology and Animal Ecology, Faculty of Agricultural and Environmental Sciences, Szent István University, H-2100, Gödöllő, Hungary

**Keywords:** neem leaf extracts, azadirachtin, *Azadirachta indica*, *Phasmarhabditis hermaphrodita*, nematodes, toxicity, biological control agents, botanical pesticides

## Abstract

Lethal effects of neem derived pesticides (neem leaf extract (NLE) and NeemAzal-T/S (NA)) were examined on different entomopathogenic (EPN) and slug-parasitic (SPN) nematodes. In our recent study, neem derived pesticides were tested against *Phasmarhabditis hermaphrodita* for the first time under in vitro conditions. Laboratory experiments were set up in 96-well microplates with different concentrations of NLE (0.1%, 0.3%, 0.6%, and 1%) and NA (0.001%, 0.003%, 0.006%, and 0.01%) and Milli-Q water as the control. After 24-h exposure time, mortality of individual nematodes was observed and recorded. Considering LC_10_ values, 0.1% of NLE could be used safely in combination with all the EPNs and SPNs tested in recent study. A concentration of NA three times higher than the recommended dosage did not harm either EPN or SPN species. In conclusion, NeemAzal-T/S might be applied with EPNs and the SPN *Ph. hermaphrodita* simultaneously, while the compatibility of neem leaf extract and beneficial nematode products needs further evaluation.

## 1. Introduction

Synthetic and natural pesticides are used in agricultural fields for crop protection, but natural pesticides are known to be eco-friendly and safe to non-target organisms [1]. *Azadirachta indica* A. Juss, 1830, commonly known as neem, is widely studied and has been used as a botanical insecticide in past decades [2]. The compound azadirachtin is known to affect the behavior and insect physiology of many plant pest species [3]. This active compound has a wide range of effects such as being an antifeedant, a disruptor of insect development, a sterilant, and causing the reduction of fitness in insects [4].

Different parts of the neem plant have a nematicidal effect on several plant parasitic nematode species such as *Meloidogyne* spp., *Rotylenchulus reniformis* Linford and Oliveira, 1940, *Pratylenchus brachyurus* Godfrey, 1929, Filipjev & Schuurmans-Stekhoven, 1941, *Hoplolaimus indicus* Sher, 1963 [5], *Globodera rostochiensis* Wollenweber, 1923, Skarbilovich, 1959 [6], and *Heterodera glycines* Ichinohe, 1952 [7]. Neem leaves and neem oil cake can cause immobility and mortality in the case of the second stage juveniles of *Meloidogyne javanica* (Treub, 1885) Chitwood, 1949, but 0.1% azadirachtin did not influence hatching [8]. On the other hand, NeemAzal-U (17% azadirachtin) could decrease hatching and viability of *Meloidogyne incognita* (Kofoid & White, 1919) Chitwood, 1949 [9]. Neem leaf extract reduced root galling caused by *M. incognita* [10]. As a side-effect, azadirachtin (0.03% Achook) can reduce the number of free living, non-target nematodes as well [11]. In order to protect non-target and beneficial organisms [12], it is a cornerstone to know sub-lethal and low-lethal concentrations of pesticides.

Entomopathogenic (Heterorhabditidae, Steinernematidae) nematodes (EPNs) have lethal effects on several pest insects [13]. Moreover, they can induce systemic resistance in plants against plant parasitic nematodes by their presence [14]. Slug-parasitic nematode (SPN) *Phasmarhabditis hermaphrodita* A. Schneider, 1859 can attack the members of Arionidae, Milacidae [15], and Limacidae families [16]. These organisms are eco-friendly; they are hard to over apply, are not harmful to humans or wildlife [16], and are compatible with numerous biological and chemical pesticides [17].

Application of biological control agents is an environmentally accepted method of pest control, but some of the pest species are becoming resistant to a single biological control agent, and therefore a combination of biological control agents must be used [18]. Recently, one of the combinations is gaining attention, as an example of integrated pest management (IPM), in the use of botanical pesticides with EPNs. A synergistic effect between azadirachtin-based botanical insecticides and the EPN was noticed whilst working on the control of peach fruit fly (*Bactrocera zonata* Saunders, 1841) [2]. The direct effects of NeemAzal-U on *Heterorhabditis bacteriophora* Poinar, 1975 was investigated [9]. Compared to the control, the highest concentration of NeemAzal-U caused significant mortality, but there was no difference in the virulence of *H. bacteriophora*. Lethal effects of Margosan-O, a commercial product of neem seed extract, on steinernematids was noticed: *Steinerenema feltiae* Filipjev, 1934 seemed more susceptible than *Steinernema carpocapsae* Weiser, 1955 or *Steinernema glaseri* Steiner,1929 [18]. 

An experiment was conducted in order to check the compatibility of soil-applied azadirachtin and *S. carpocapsae* to control western flower thrips (*Frankliniella occidentalis* Pergande, 1985). EPN combined with NeemAzal-T (1%) or with neem pellets achieved an additive effect on *F. occidentalis* [3]. Different neem pesticides were tested on the viability and virulence of *S. feltiae*. Neem products did not have any negative effect on EPN virulence or viability [19]. *S. feltiae* and *H. bacteriophora* compatibility with azadirachtin was proven (with 0% mortality) [20]. Although the efficacy of neem has already been examined on pest slugs [21] and snails [22], the interaction between neem derived pesticides (either traditional or commercial) and *Ph. hermaphrodita* has not been studied and documented yet.

This is the first time that investigation of the side-effects of neem derived pesticides on the SPN, *Ph. hermaphrodita* has been conducted. Our further aim was to compare different concentrations of neem leaf extract (NLE) and NeemAzal-T/S (NA) on the commercially available EPN and SPN species in Hungary.

## 2. Results

In the case of *H. bacteriophora*, a steep slope could be noticed; 0.1% NLE did not cause any lethal effect, while 0.3% NLE resulted in 97.5% mortality (Figure 1A). NLE of 0.3% concentration did not cause any effect on the viability of *Ph. hermaphrodita* juveniles. Only at higher concentrations (i.e., 0.6 and 1%), could 85% and 95% mortality be observed, respectively (Figure 2A).

Only 13.75% of *S. carpocapsae* juveniles died by 0.3% NLE, while 0.6% and 1% NLE caused 80.36% and 79.64% mortality, respectively (Figure 3A).

In the case of *S. feltiae*, there was a slight stepwise increase but no significant difference between mortality at control, 0.1%, and 0.3% NLE, with the highest average of mortality being 19.4%. Efficacy of 0.6 and 1% NLE concentrations showed between 70.5% and 90.8% mortality, respectively (Figure 4A). NLE of 0.1% did not have any effect on the survival of *Steinernema kraussei* Steiner, 1923 juveniles, while 0.3% NLE caused 46.5% mortality, whereas 95% and 100% mortality was observed in higher (0.6% and 1%) concentrations, respectively (Figure 5A).

In the case of NA concentrations, mortality results of every species were inconsistent (Figure 1B, 2B, 3B, 4B, and 5B). Considering all species and every concentration, the highest mortality was only 15.6% in the case of *S. carpocapsae* at 0.003% NA concentration (Figure 3B).

Since the mortality was inconsistent at every concentration of NA, with even the highest values being much lower than 100%, LC values were not calculated.

On the other hand, values of NLE LC_50_ were determined for the different nematode species. *H. bacteriophora* had the lowest value (0.217%), while *S. feltiae* had the highest (0.480%). In *Ph. hermaphrodita*, NLE concentration of 0.366% caused the mortality of 50% of juveniles. In the case of LC_10_ concentrations, a different tendency developed. *S. carpocapsae* had the highest value (0.293%), followed by *S. kraussei* (0.185%), *H. bacteriophora* (0.179%), *S. feltiae* (0.172%), and *Ph. hermaphrodita* (0.132%) (Table 1).

## 3. Discussion

There was species-specific variation in the response of the nematodes tested with the various concentrations of NLE, which was a similar finding when compared with the previous study in which EPN species could have different sensitivity against fungicides [23]. Considering LC_50_ values, *H. bacteriophora* seemed the most sensitive, which is in accordance with a previous study [24]. *Ph. hermaphrodita* LC_50_ was the second highest (0.366%), and its LC_10_ value was the lowest (0.132%) from all species.

Neem leaf extract had a stronger lethal effect than NeemAzal T/S on the examined nematode species. One of the possible reasons could be that leaf extract contains higher azadirachtin content than the commercial product. Another possible explanation could be that NLE does not consist only azadirachtin but other pesticidal active compounds known as ‘triterpene’, more specifically ‘limnoids’, e.g., nimbin, nimbidine, nimbinin, azadirachtol, salannin, and other such derivatives, which may exhibit a toxic effect [1]. According to a previous study, aqueous extract of neem leaf had higher salannin content than azadirachtin content [25]. In addition, both nimbin and salannin have a nematicidal effect [26,27]. Moreover, compounds found in neem extracts from different parts of the neem tree may enhance the effect of each other by synergism [2,3,9,19,28].

Considering LC_10_ values, 0.1% of NLE might be used safely in combination with each examined EPN and SPN species. Since the mortality results of NA were inconsistent and low, the low- and sub-lethal values could not be calculated. However, we demonstrated under in vitro conditions that concentrations of NA three times higher than the recommended for field applications and neem leaf extract in low concentration did not harm either EPN or SPN species. As a conclusion, the commercial neem product may be applied with EPNs and/or *Ph. hermaphrodita* simultaneously as plant protection agents, although further research regarding its field application needs to be investigated. Furthermore, compatibility of neem leaf extract and beneficial nematode species also requires further evaluation.

## 4. Materials and Methods

### 4.1. Preparation of Neem Leaf Extract (NLE)

The methodology followed a similar study [29], with certain modifications. Pre-air-dried neem leaves obtained from India were ground to powder by a blender, and a stock concentration of 5% (w/v) solution using distilled water was prepared and kept in the dark at room temperature for 24 hours. The stock solution was filtered the next day using a muslin cloth to obtain clear water extract. Further, it was centrifuged at 5000 rpm for 10 min to remove particulate matter. A maximum concentration of 1% of NLE was chosen, as a preliminary pilot study performed with 1% and 5% NLE resulted in 100% mortality of all tested nematode species. Therefore, range of concentration of NLE for the treatment was 0.1%, 0.3%, 0.6%, and 1%.

### 4.2. Preparation of Azadirachtin (NA)

NeemAzal-T/S (Trifolio-M GmBH) is a commercial product registered in the European Union, containing 1% azadirachtin. According to the Hungarian approval document of NA, the maximum azadirachtin concentration of the applied spray mixture could be 0.003% against glasshouse whitefly (*Trialeurodes vaporariorum* Westwood 1856) in protected tomato (04.2/4878-1/2012. Nébih 2018) [30]. In order to simulate overdosage, we examined concentrations of NA ranging between 0.001% and 0.01%, prepared by diluting the original product with distilled water.

### 4.3. Entomopathogenic (EPN) and Slug-Parasitic (SPN) Nematodes

The EPN species (products of Biobest, Belgium) used in the experiments were *Heterorhabditis bacteriophora* (B-Green), *Steinernema carpocapsae* (Carpocapsae-System), *Steinernema feltiae* (Steinernema-System), *Steinernema kraussei* (Kraussei-System)*,* and the SPN species (product of Biobest, Westerlo, Belgium) was *Phasmarhabditis hermaphrodita* (Phasmarhabditis-System). All products were stored in the refrigerator at 5 °C until they were used for the experiment.

### 4.4. Experimental Setup

The experiment was performed in vitro in flat-bottomed, 96-well microplates (Kartell S.p.A., Noviglio, Italy). Five juveniles from each nematode species were placed into each well with 60 µL distilled water using a micropipette. That was followed by treatment with the addition of 200 µL of each concentration and 200 µL of distilled water for control. Eight replicates of each treatment were applied. The microplates were closed with lids and sealed by parafilm tape to avoid evaporation of the extracts and incubated in a thermostat in dark conditions at 20 °C ± 1 °C. The plates were checked after an exposure period of 24 h under a transmission stereomicroscope. Nematode viability was evaluated based on movements assessed by adding 10 µL of 5% lactic acid, a modified method [31] in which 4% lactic acid was used as a movement stimulant. Maximum mortality of 20% in the control treatment was considered as a validity criterion for the tests [32].

### 4.5. Data Elaboration and Statistical Analysis

Data were processed and square root arcsine-transformed in an Excel spreadsheet before statistical analysis using PAST3 (Paleontological Statistics) statistical software [33]. One-way ANOVA, more specifically Tukey’s test and Mann–Whitney U test, were performed on the data, depending on whether the normality (Shapiro–Wilk test) was fulfilled. Graphs were made using the MS Excel 2016 program. Low-lethal (LC_10_) and sublethal (LC_50_) values for different NLE concentrations were calculated by AAT Bioquest^®^ calculator.

## Figures and Tables

**Figure 1 plants-08-00281-f001:**
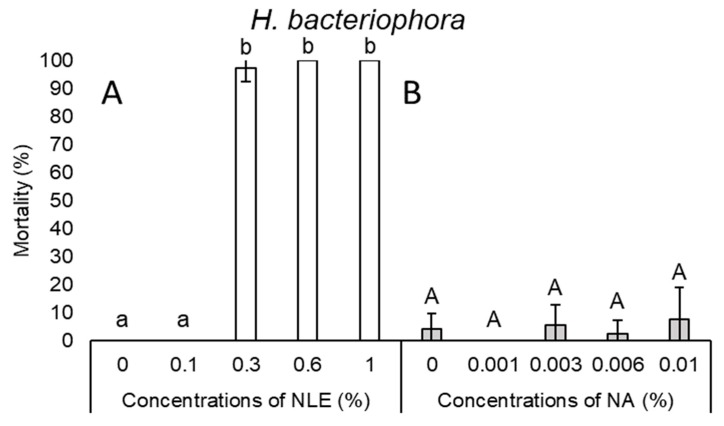
Mortality after 24-h exposure time of *Heterorhabditis bacteriophora* juveniles of 0.1%, 0.3%, 0.6%, and 1% neem leaf extract (NLE) (**A**) and 0.001%, 0.003%, 0.006%, and 0.01% of NeemAzal-T/S (NA) (**B**). (One-way ANOVA, Mann–Whitney U test; the same letters indicate no significant difference at *p* < 0.05 level).

**Figure 2 plants-08-00281-f002:**
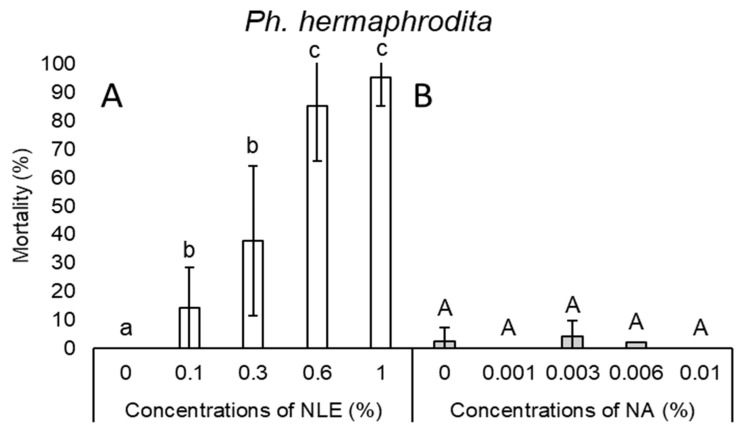
Mortality after 24-h exposure time of *Phasmarhabditis hermaphrodita* juveniles of 0.1%, 0.3%, 0.6%, and 1% NLE (**A**) and 0.001%, 0.003%, 0.006%, and 0.01% of NA (**B**). (One-way ANOVA, Mann–Whitney U test; the same letters indicate no significant difference at *p* < 0.05 level).

**Figure 3 plants-08-00281-f003:**
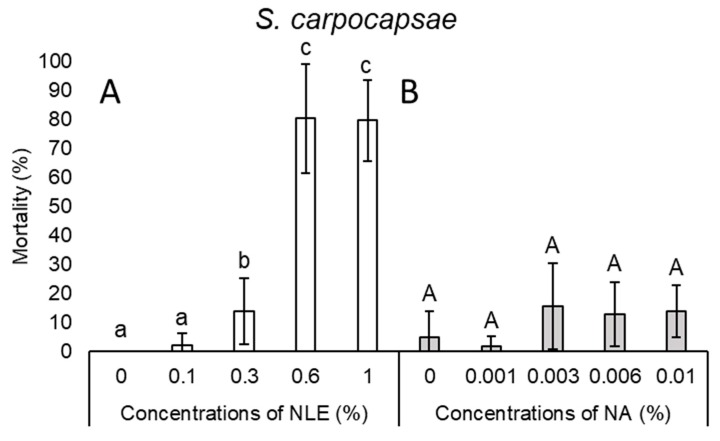
Mortality after 24-h exposure time of *Steinernema carpocapsae* juveniles of 0.1%, 0.3%, 0.6%, and 1% NLE (**A**) and 0.001%, 0.003%, 0.006%, and 0.01% of NA (**B**). (One-way ANOVA, Mann–Whitney U test (**A**), Tukey’s pairwise comparisons (**B**); the same letters indicate no significant difference at *p* < 0.05 level).

**Figure 4 plants-08-00281-f004:**
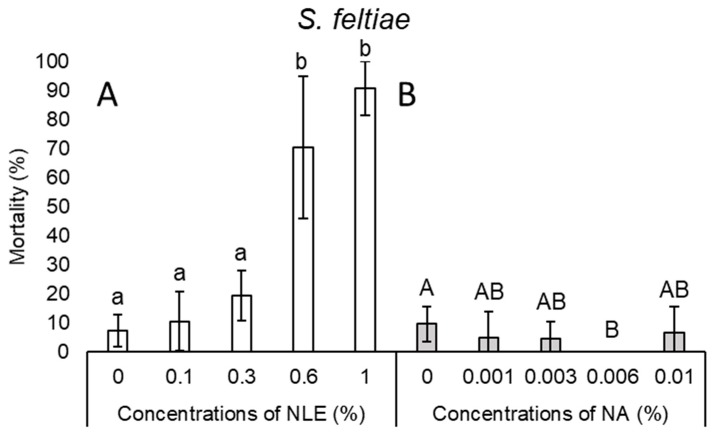
Mortality after 24-h exposure time of *Steinernema feltiae* juveniles of 0.1%, 0.3%, 0.6%, and 1% NLE (**A**) and 0.001%, 0.003%, 0.006%, and 0.01% of NA (**B**). (One-way ANOVA, Tukey’s pairwise comparisons (**A**), Mann–Whitney U test (**B**); the same letters indicate no significant difference at *p* < 0.05 level).

**Figure 5 plants-08-00281-f005:**
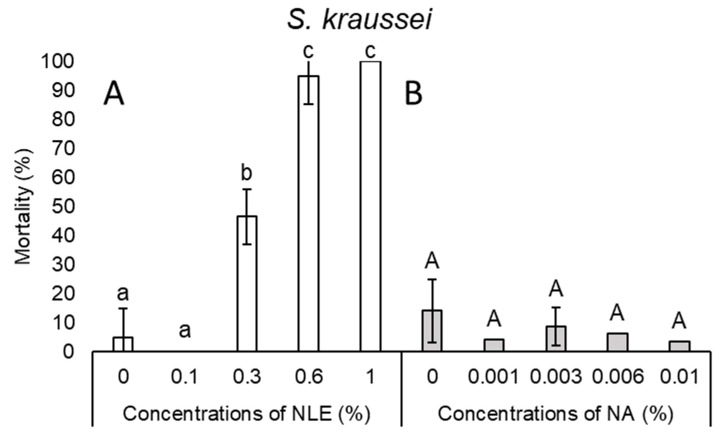
Mortality after 24-h exposure time of *Steinernema kraussei* juveniles of 0.1%, 0.3%, 0.6%, and 1% NLE (**A**) and 0.001%, 0.003%, 0.006%, and 0.01% of NA (**B**). (One-way ANOVA, Mann–Whitney U test (**A**), Tukey’s pairwise comparisons (**B**); the same letters indicate no significant difference at *p* < 0.05 level).

**Table 1 plants-08-00281-t001:** LC_10_ and LC_50_ values for NLE (%) in the cases of Heterorhabditis bacteriophora, Phasmarhabditis hermaphrodita, Steinernema carpocapsae, Steinernema feltiae, and Steinernema kraussei.

Species	LC_10_	LC_50_
*H. bacteriophora*	0.179%	0.217%
*Ph. hermaphrodita*	0.132%	0.366%
*S. carpocapsae*	0.293%	0.330%
*S. feltiae*	0.172%	0.480%
*S. kraussei*	0.185%	0.313%

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
