# Peer review of "Investigating the Side-Effects of Neem-Derived Pesticides on Commercial Entomopathogenic and Slug-Parasitic Nematode Products Under Laboratory Conditions"

_plants, 2019, doi:10.3390/plants8080281_

Round 1
Reviewer 1 Report
This manuscript describes an evaluation of one neem-derived commercial product used for insect pest control, as well as, a laboratory made neem extract on beneficial entomopathogenic nematodes, as well as, nematodes that serve as predator/parasites of crop damaging slugs. The overall purpose was to determine whether these neem-derived products would affect beneficial nematodes in a detrimental way to determine if these two forces could be used in combination for insect pest control. The toxicity of the neem products against the nematode species was determined. All concentrations of the commercial product did not affect nematodes species negatively as compared with the control. Furthermore, at concentrations below 0.6%, the neem leaf extract created was not lethal to either nematode species. The overall conclusion of the investigation is that need-derived products should be compatible with beneficial nematode formulations as a combined method of insect pest control. The investigation is small and the topic is somewhat limited in scope; however, it is useful to know that neem-based products could be compatible with beneficial arthropod or slug-infesting nematodes as combined methods of control. It would be useful to conduct field testing in the future as a follow up and further prove that the combination of neem + nematodes has additive or synergistic effects with respect to reducing pest populations.
The manuscript is generally well written; however, I have some editorial comments below. The manuscript would likely use more editing for English grammar. Overall, I think the methods were sufficient to test the hypotheses proposed. The replication and statistical analyses seem adequate. The results are pretty clear cut and the conclusions seem to follow from the results obtained.
Line 22: “In conclusion”
Line 40: “can cause”
Line 44: “can reduce”
Line 70: “Neem products did not have any…”
Line 71: “…was proven.”
Line 76: “…was conducted.”
Line 81: Delete “already”
Line 91: Italicize genus species name.
Line 131: “There was species-specific variation in the response of the nematodes tested to the various concentrations of…”
Lines 153-154: “…needs to be investigated. Furthermore, compatibility of…also requires further evaluation.”
Line 157: Delete: “was like”
Line 171-172: “…, we examined concentrations of NA ranging between 0.001 and 0.01%, prepared…”
Author Response
Dear Reviewer 1,
Thank you for sparing your time and giving us your comments and suggestions. We made the following changes here in italics and using track changes in the manuscript:
Line 22: “In conclusion” inserted in line 21
Line 40: “can cause” inserted in line 38
Line 44: “can reduce” inserted in line 43
Line 70: “Neem products did not have any…” inserted in line 66
Line 71: “…was proven.” inserted in line 68
Line 76: “…was conducted.” inserted in line 73
Line 81: Delete “already” deleted in line 78
Line 91: Italicize genus species name. changes made in line 87
Line 131: “There was species-specific variation in the response of the nematodes tested to the various concentrations of…” changes made in line 125,126
Lines 153-154: “…needs to be investigated. Furthermore, compatibility of…also requires further evaluation.” changes made in line 147, 148
Line 157: Delete: “was like” changes made in line 151
Line 171-172: “…, we examined concentrations of NA ranging between 0.001 and 0.01%, prepared…” changes made in line 164,165
-------------------------------------------------------------------------------------
We also made some changes in the formatting according to the Plants guidelines template. Below are the following changes:
The entire text was changed to font Palatino Linotype, 10 pt
2. Fig was replaced by 'Figure' for all the graphs and for respective text

Reviewer 2 Report
In this paper authors investigated the effects of two Neem extracts on different entomopathogenic (EPN) and slug parasitic (SPN) nematodes. For the purpose the used a leaf extract and a commercial product.
I think this manuscript is well written and scientifically sound. However, I found some points to be improved.
Please give more insights into the phytochemical composition of Neem leaf extract. Report HPLC chromatogram and/or list of extract constituents and their content. That is very important because this extracts was more active than the commercial one.
Other minor changes:
- Insert authorship at first citation of scientific names.
- Do not abbreviate 'S.' at first citation. What does 'S.' mean? Specify the genus.
- Fig. 1: above some bars there is no SD range.
- Fig. 2. Phasmarhabditis hermaphrodita should be in Italics.
- Sect. 4.1 Should be Preparation 'of' neem leaf extract.
Author Response
Dear Reviewer 2,
Thank you very much for sparing your time and giving your comments and suggestions on our manuscript. We made the changes in italics here and using track changes/comment in the manuscript. The changes/corrections are as follows:
Please give more insights into the phytochemical composition of Neem leaf extract. Report HPLC chromatogram and/or list of extract constituents and their content. That is very important because this extracts was more active than the commercial one.
Yes we agree with your comments that the extract was more active than the commercial product. Therefore, we plan to make a quantitative and qualitative chemical analysis of neem leaf extract in the future, and we are going to test all those compounds separately that might have biological activity on beneficial nematodes and root-knot nematodes as well.
Also we insert text in the manuscript regarding the contents of neem in line 132 - 136 highlighted in yellow.
Other minor changes:
- Insert authorship at first citation of scientific names. inserted authorship and year at first citation of all the scientific names
- Do not abbreviate 'S.' at first citation. What does 'S.' mean? Specify the genus. It has been changed to the Genus name Steinernema in lines 61 and 62 and species first letters are changed to lowercase
- Fig. 1: above some bars there is no SD range. inserted comment for fig 1.
- Fig. 2. Phasmarhabditis hermaphrodita should be in Italics. changed to italics in line 87
- Sect. 4.1 Should be Preparation 'of' neem leaf extract. changes made in line 150
---------------------------------------------------------------------------------------
We also made other changes in the manuscript which are as follows:
Entire text formatted as per guidelines of Plants journal i.e font Palatino Linotype, 10 pt. References changed according to the guidelines of Plants journal. Inserted the word 'neem' oil cake.... in line 38
